# The Impact of SARS-CoV-2 Pandemic Lockdown on a Botulinum Toxin Outpatient Clinic in Germany

**DOI:** 10.3390/toxins13020101

**Published:** 2021-01-29

**Authors:** Sara Samadzadeh, Raphaela Brauns, Dietmar Rosenthal, Harald Hefter

**Affiliations:** Department of Neurology, University Hospital of Düsseldorf, D-40225 Düsseldorf, Germany; sara.samadzadeh@yahoo.com (S.S.); raphaela.brauns@uni-duesseldorf.de (R.B.); rosenthal@med.uni-duesseldorf.de (D.R.)

**Keywords:** SARS-CoV-2 virus, COVID-19 pandemic, lockdown, delay in treatment, worsening of symptoms, cervical dystonia

## Abstract

Botulinum neurotoxin type A (BoNT/A) injections have to be administered repeatedly to achieve a rather stable, high level of improvement. This study aimed to take a look at changes in the daily routine of a BoNT/A outpatient clinic due to the SARS-CoV-2 pandemic lockdown, analyze the impact of SARS-CoV-2-induced re-injection delay on outcomes in patients with cervical dystonia (CD) (*n* = 36) and four other disease entities (*n* = 58), and study the influence of covariables, including previous injections and doses. For the present observational study, the first 100 patients who were scheduled to have an appointment between April 20 and May 18 during the partial lockdown and also had been treated regularly before the lockdown were recruited. Clinical and demographical characteristics and treatment-related data from the previous visits were extracted from charts. Time delay, symptom severity assessment, and TSUI score (if applicable) were gathered at the first coronavirus pandemic lockdown emergency visit for each patient. Of the 94 patients who could come to the clinic, 48 reported a delay and 44 reported worsening during the delay. Delays ranged from 1 to 63 days, the mean delay was 23 days, and the mean worsening was 26% compared to the previous visit. A significant correlation was found between the duration of the delay and the patient’s rating of worsening (PwP). In CD patients, the physician´s rating of CD worsening by the TSUI score (ATUSI-PTSUI) was significantly correlated with general worsening (DwP) and the TSUI at the last visit (PTSUI). A small delay of a few weeks led to a similar worsening of symptoms in CD and all other disease entities and to relapse on a higher level of severity. This relapse can only be compensated by continuous treatment up to at least 1 year until patients reach the same level of treatment efficacy as that before the SARS-CoV-2 pandemic.

## 1. Introduction

Botulinum neurotoxin type A injections have become the treatment of choice for a variety of disease entities [1]. For most of these indications, injections have to be administered repeatedly to achieve a rather stable, high level of improvement [1,2]. The injection cycle and interval are constant in our department [3]. About 80% of the patients are re-injected after 12 to 13 weeks [4] and, if possible, at the same time of the day.

The SARS-CoV-2 pandemic reached Germany at the end of January 2020, and a hot spot gradually initiated and then developed close to Düsseldorf after the carnival [5]. The transport of COVID-19-infected patients to the University Hospital of Düsseldorf (UKD) led to the lockdown of all outpatient clinics in UKD from the 15th of March and a ban to visit patients except in emergency cases from the 10th of February.

This early lockdown heavily interfered with patient management in our clinic. Due to the extension of the lockdown, the number of patients who called for a new appointment and reported that their symptoms had considerably worsened increased and an emergency situation developed. Finally, we organized special appointments and time slots for emergency cases on the 15th of April, but it was quite unexpected to see that only a few patients presented with a constellation of symptoms deserving to be in the emergency classification, such as functional blindness in patients with blepharospasm.

When the general lockdown ended on the 20th of April 2020, the clinic started again under careful control of hygiene standards. The SARS-CoV-2-induced re-injection delay varied considerably. This provided a unique chance to analyze the impact of this delay on outcomes in our outpatient clinic and to study the influence of covariables such as the dose of the preceding injection.

## 2. Results

In the time period between the 20th of April and the 18th of May, 100 patients had appointments for BoNT/A therapy. Four patients from outside of Germany (one from Greece, one from Spain, and two from Belgium) had to postpone their therapies again because Germany had kept its borders closed for people from those countries. Permission for treatment in Germany for a fifth patient from Spain with generalized dystonia did not arrive on time. One other patient, an 82-year-old male with left lower spasticity after a stroke, survived COVID-19 and is still in a special rehabilitation program; therefore, his BoNT/A treatment is further postponed.

Of these 94 patients, only a 42-year-old male patient with right-sided hemiparesis after left carotid artery dissection who had been on holidays in the hot spot Ischgl (Austria) became infected. After a few days with mild fever and coughing, he recovered. He did not experience a relevant worsening of spasticity (5%) during the 14-day delay. The patient with the longest delay of 63 days and the maximum worsening of 60% lived in the hot spot district of Heinsberg (close to Düsseldorf, Germany). He had been in quarantine weeks before the lockdown in our institution started. He suffered from severe blepharospasm but responded well to BoNT/A therapy, so he was able to drive a car. During the 63 days of delay, he developed episodes of several minutes’ length of functional blindness.

In Figure 1a, the assessment of the efficacy of the first seven cycles of BoNT/A treatment of a 49-year-old male with idiopathic CD is presented. The eighth injection had to be postponed for 4 weeks. During this time, the patient experienced worsening of 30% and relapse to a severity level of 1 year before. This patient claimed to be an emergency case.

In Table 1, demographical data and treatment-related data of all 94 patients and 5 disease entities are presented. One patient was injected every 15 weeks, 63 patients every 13 weeks, 12 patients every 12 weeks, 9 patients every 11 weeks, 6 patients every 10 weeks, 2 patients every 9 weeks, and 1 patient every 8 weeks.

Of the 94 patients, 48 reported a delay and 44 reported worsening during the delay. Delays ranged from 1 to 63 days, the mean delay was 23 days, and the mean worsening was 26% in patients with a delay who noticed worsening. Only four patients with a delay did not experience worsening during the delay. Furthermore, four patients who did not have a delay reported worsening.

Different reasons for an explanation of the worsening were offered by the patients. In most cases, a reduction in the efficacy of the previous BoNT/A injection was experienced. But in three cases, cessation of physiotherapy or an increase in psychosocial stress because of additional nursing of a relative who had been in an outpatient care unit before the lockdown (one case) was mentioned.

A highly significant non-parametric correlation (Spearman´s rho correlation coefficient *r* = 0.84; *p* < 0.001) was found between the rating of worsening by the patient and the treating physician. A significant non-parametric correlation (*r* = 0.66; *p* < 0.001) was found between the duration of the delay and the patient´s rating of worsening. The treating physician´s rating correlated slightly higher (*r* = 0.68; *p* < 0.001) with the delay than the patient´s rating. The physician´s rating tended to be non-significantly lower than the patient´s rating (Table 1).

The Pearson correlation between worsening (in %) and delay (in days) in CD patients (y = 0.6x + 5; *r* = 0.48; *p* < 0.05) was almost identical to the correlation between worsening and delay in non-CD patients (*r* = 0.54; *p* < 0.01; Figure 1b). This was true for both patients’ and the treating physician´s rating.

The physician´s rating of CD worsening by the TSUI score (ATUSI–PTSUI) did not correlate with the delay but was significantly correlated with the physician´s rating of the percentage of worsening (*r* = 0.43; *p* < 0.01) and was significantly correlated with the PTSUI score (*r* = 0.41; *p* < 0.01). There was a highly significant correlation between the TSUI score determined by the treating physician and the independent rater (ITSUI score; *r* = 0.91; *p* < 0.001).

The stepwise regression analysis did not reveal any significant influence on the worsening other than the delay. Neither the disease entity nor the dose of the previous treatment was included as a second factor.

## 3. Discussion

In our BoNT/A outpatient clinic, most patients are treated with fixed-cycle durations, mostly 12 or 13 weeks. An increase in the inter-injection interval leads to a reduction in efficacy at the end of the treatment cycle. This is demonstrated in Figure 1a for a patient with CD but was observed in all five disease entities in this study group (Table 1). On average, one day of delay caused 1% worsening. During a delay of 3 to 4 weeks, worsening approached a clinically relevant level of about 25% (see Table 1 and Figure 1a,b).

In general, the duration of efficacy of a single BoNT/A injection exceeds 12 to 13 weeks in many patients. Even after 16 weeks, a significant effect can be demonstrated for ona- and incoBoNT/A injections [6]. Re-injections after 12 to 13 weeks before the effect of the previous injection has fully declined will lead to a staircase-like continuous improvement injection by injection until a rather stable level of improvement is reached (see Figure 1a). This improvement is about 60% of the initial severity of symptoms, is much larger than the peak effect at week 4 of the first injection (see Figure 1a), and leads to high patient satisfaction with BoNT/A therapy [7,8].

There was a significant (*p* < 0.01) correlation between the percentage of worsening of symptoms assessed by the patients and the treating physician. The treating physician also determined the TSUI score, but compared to the percentage scale, the TSUI score is less sensitive at picking up mild changes in the severity of CD.

The duration of efficacy during BoNT/A therapy usually depends on the total dose, the concentration of BoNT/A injected, the type of nerve terminal blocked (motor, sensory, peptidergic, cholinergic), and the recovery rate of SNAP-25 cleaved by BoNT/A [9]. A similar amount of worsening independent on these factors including dose after prolongation of the treatment cycle across all disease entities indicates that the amount of BoNT/A is individually adapted to the duration of the treatment cycle.

Patients being treated with variable inter-injection intervals between 9 and 16 weeks will be used to the resulting fluctuations and therewith will be less sensitive to the experience of worsening when the injection interval is prolonged up to 4 weeks once. It may very well be that patients in our BoNT/A clinic were highly sensitized to a postponement of the re-injection. The rating of the treating physician was less pronounced compared to the patient´s rating. Usually, treating physicians overestimate a treatment effect [7].

The reported experience of the impact of the delay caused by the first lockdown had immediate consequences for patient management during the second lockdown, which started in Germany on the 16th of December 2020. On the basis of the present data, we applied for an exceptional permission not to close the BoNT/A outpatient department but to continue BoNT/A injection therapy for emergency cases such as the patient with functional blindness mentioned in Section 2 of the Results.

## 4. Conclusions

The present observation of worsening of symptoms with prolongation of the treatment cycle due to the SARS-CoV-2 pandemic can be interpreted as an argument for rigid patient management with fixed treatment cycles. Re-injection before the efficacy of the previous injection has completely vanished can lead to a staircase-like continuous improvement and high patient satisfaction and a reduction in the burden on the BoNT/A-clinic.

The SARS-CoV-2 pandemic heavily interfered with this rigid patient management and gave us a unique opportunity to study the impact of a systematic delay on the outcome in our clinic. A small delay of a few weeks can, therefore, lead to worsening of symptoms and relapse on a level of severity, which therefore may take several treatment cycles to return to the prior stable benefit level before the SARS-CoV-2 pandemic. The experience of this emergency situation is from the patients’ view very well taken.

Therefore, BoNT/A outpatient departments should not be shut down, even during a general lockdown in hospitals.

## 5. Materials and Methods

The botulinum neurotoxin type A (BoNT/A) clinic of UKD closed from the 15th of March and opened again on the 20th of April. The first 100 patients who were scheduled to have an appointment between the 20th of April and the 18th of May during the partial lockdown and who had been treated on a regular basis before the lockdown were recruited for this observational study. (This study has been approved by the local ethics committee of the University of Duesseldorf (number: 4085, First date: 5 April 2013, Second updated date: 8 May 2018)).

Patient history since the last injection was explored by the treating physician (HH). He asked whether this appointment with the clinic had been postponed because of the SARS-CoV-2 pandemic and whether a patient had experienced worsening of symptoms during this delay, and he rated the worsening as a percentage of disease severity at the onset of BoNT/A therapy. Finally, the patients had to rate the worsening by marking the actual severity of the disease on a visual analogue scale ranging from 0% (no symptoms) to 100% (severity of symptoms at therapy onset). On the day of recruitment, the therapy was continued as before the lockdown.

The patients’ age, sex, date of last injection, preparation used for the last injection, and last dose were extracted from charts. To compare doses of the three BoNT/A preparations, the doses were converted to unified dose units (uDU) by leaving onaBoNT/A (Botox^®^) and incoBoNT/A (Xeomin^®^) doses unchanged and by dividing aboBoNT/A (Dysport^®^) doses by a factor of 3 following a European consensus paper [10]. Five disease entities were distinguished: Check whether edits retain the intended meaning.blepharospasm and facial hemispasm), idiopathic cervical dystonia (CD), other dystonia (ODT such as oromandibular, oropharyngeal, trunk, and extremity dystonia), spasticity after infarct or trauma (SPAS), and PAIN (pain syndromes such as migraine or tension headache).

In patients with idiopathic CD, the TSUI score at the present visit (ATSUI; [11]) was determined by the treating physician (HH) and by an independent physician (RB) not familiar with the treatment data of the patients (ITSUI). The TSUI score determined at the previous injection visit (PTSUI) was extracted from the charts.

A stepwise binary regression analysis was calculated to determine whether worsening of symptoms was influenced by the delay, a previous dose of BoNT/A, or a disease entity. Furthermore, non-parametric rank correlation analysis was performed. Determination of regression lines and use of Pearson correlation analysis were explicitly mentioned. All statistical tests were part of the SPSS statistics package (version 25, IBM, Armonk, NY, USA).

## Figures and Tables

**Figure 1 toxins-13-00101-f001:**
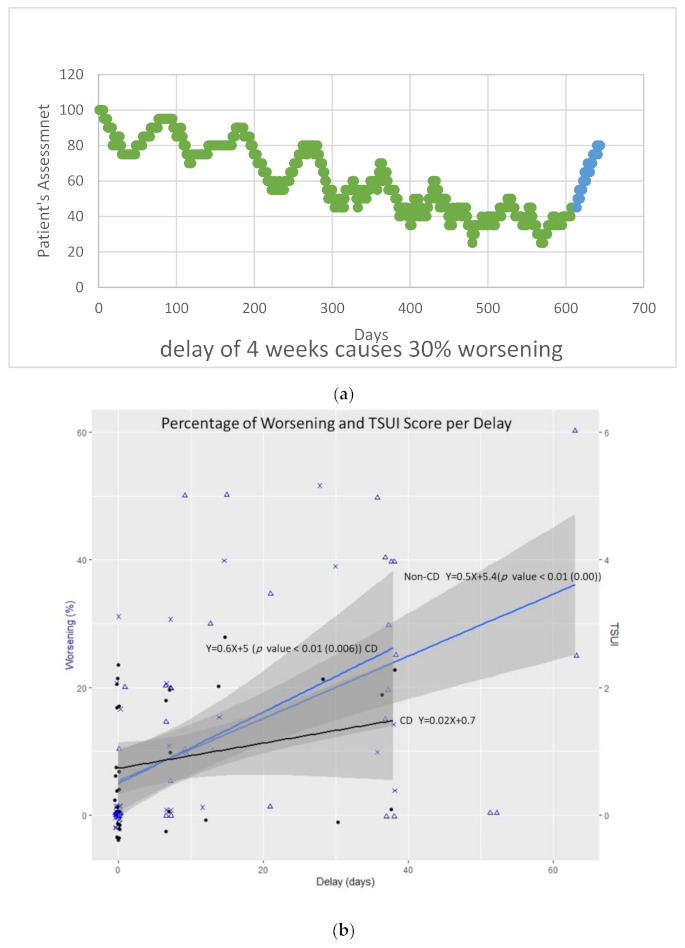
(**a**) One patient’s assessment of disease severity during seven cycles before the lockdown (green circles) and worsening during the lockdown (blue circles) as an example of the delay’s effect on worsening. (**b**) Linear regression model between the worsening of the disease and the delay in re-injection in cervical dystonia (CD) patients (CD) and non-CD patients. The TSUI score is insensitive at picking up mild changes in the severity of CD, but the treating physician´s rating of worsening and the change in the TSUI score were significantly (*p* < 0.01) correlated.

**Table 1 toxins-13-00101-t001:** Demographical and treatment-related data in 5 disease entities.

	Disease Type	FD	CD	ODT	SPAS	PAIN	ALL
Parameter							
**No. Patients**		25	36	15	10	8	94
**Age** **(Mean+/−SD)**		70+/−9	67+/−12	51+/−15	64+/−15	53+/−10	64+/−14
**No. of Patients** **with delayed injections**		13	16	9	7	3	48
**No. of Patients** **with Worsening symptoms**		10	17	8	6	3	44
**Time Delay (days)** **(Mean+/−SD)**		21+/−25	21+/−12	25+/−18	31+/−16	19	23+/−16
**Percentage of Worsening** **(by patients) (PwP)** **(Mean+/−SD)**		33+/−19	22+/−12	24+/−10	23+/−9	28	26+/−14
**Percentage of Worsening** **(by physician) (DwP)** **(Mean+/−SD)**		23+/−17	21+/−9	19+/−5	18+/−7	22	21+/−11
**Botulinum Toxin Dose(uDU)** **(Mean+/−SD) ***		63+/−47	295+/−138	153+/−111	326+/−170	133+/−58	199+/−155
**Last visit TSUI score (PTSUI)**		NA	4+/−2	NA	NA	NA	NA
**TSUI score by physician (ATSUI)**		NA	5+/−3	NA	NA	NA	NA
**TSUI score by independent rater** **(ITSUI)**		NA	5+/−2	NA	NA	NA	NA

* To compare doses of the three BoNT/A preparations, doses were converted to unified dose units (uDU) by leaving onaBoNT/A (Botox^®^) and incoBoNT/A (Xeomin^®^) doses unchanged and by dividing aboBoNT/A (Dysport^®^) doses by a factor of 3.

## Data Availability

Data available on request due to restrictions eg privacy or ethical. The data presented in this study are available on request from the corresponding author.

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
