# Peer review of "The Impact of SARS-CoV-2 Pandemic Lockdown on a Botulinum Toxin Outpatient Clinic in Germany"

_toxins, 2021, doi:10.3390/toxins13020101_

Round 1

Reviewer 1 Report

Reviewers report: The impact of SARS-CoV-2-pandemic lockdown on botulinum toxin outpatient clinic in Germany

This report examines the impact of the COVID-19 shut down on the administration of onabotulinumtoxin for in a cohort of 100 patients Undergoing therapy for neurological indications . The Report also describes the consequences of delayed treatment upon disease severity.

The abstract is a concise summary of the study and requires little editing

Introduction:

the introduction requires some editing for the use of English But sets the scene accurately. The sturdy question is clear

methods: This may appear a little “woke” but the physicians are described as a male perhaps sex and neutral phrases could be used?

Was the validity of the subjective scale of symptom worsening validated at all or used in previous studies showing that it was reliable and accurate?

Results

The authors present valid analysis and case illustrations the results are well written and easy to understand

discussion:

clearly the results of the study are limited by the absence of any control comparator and one wonders about the validity of such statements as one day of delay leading to a significant worsening in symptoms . To what extent do the authors consider that this may be as much as psychological affect as a disease related impact ?

undoubtedly significant delay does however leads to worsening of symptoms And the authors make very valid points about the additional time required in “catch up “ And regarding the sensitization of patients to their individualized treatment cycle

The authors make an interesting point about the difference in patients and physicians ratings over symptom worsening whereby physicians usually overestimate  (presumably in this scenario ) the impact compared to patience . There is some interesting qualitative work to be done here in many other clinical situations physicians tend to minimize symptoms when compared to patients subjective estimation

The authors suggested their findings would support a more rigid, rather than individually tailored pre treatment cycle. What is lacking here is Data regarding the extent of the additional advantage gained by re injection prior to complete abolition of affect of botulinum toxin injection . Is the additional benefit markedly noticeable by patience and would not this additional benefit be lost with introduction of a rigid treatment cycle ?

regardless, the authors have illustrated the impact of treatment delay in response to the pandemic and yet another adverse consequence 2 patients with non coronavirus disease

the authors might like to reference one or two papers which have addressed this from a different perspective in different disease entities

Author Response

This is an adequate short summary of our manuscript. But as we have mentioned in section 3 of Materials and Methods, we have used all 3 BoNT/A preparations licensed in Europe.

Language has been improved.

All patients were treated by HH (male) and also rated by RB (female). By mentioning the names, it should be clear who is male and who is female. 

The scale used was a standard visual analogue scale (VAS) and rating was performed by making a mark on the VAS. This is mentioned now in the text.

We do not say that one day of delay leads to a significant worsening in symptoms. Our statement is that the steepness of the regression line (0.6%/day) is significant. After 23 days in the mean this steepness leads to a worsening of more than 25% in the mean. This is an amount of worsening that really matters.

We think that this increase of worsening is disease independent and not a psychological affect, but has to do with reduction of efficacy of BoNT/A.

Reviewer 1 is absolutely right:

In the paper by Moll et al (2018) cited in the text it is demonstrated that rating of patients and physicians correlate quite well for motor symptoms but not for symptoms as pain, emotional well-being and impairment during daily life. The physician rates what he sees, the patient also what he feels.

We hope citation of this paper is enough to address this point, we did not repeat what has been presented previously.

This point raised by Reviewer 1 is also picked-up by other reviewers.

Usually, the effect of BoNT action is analyzed around week 4, when BoNT has its maximal effect. But in the present paper the effect is rated at the end of an injection cycle when the BoNT action declines. As demonstrated in Fig. 1

repeated injections at week 12 or 13 at a time point when the BoNT action has not fully declined lead to an improvement much higher than the improvement of symptoms at week 4.

The improvement increases step by step with each injection, but the improvement achieved by a single injection decreases with time and the number of injections. In Fig. 1 the peak effect of the first effect was about 25% improvement. The peak effect of the 7th injection was about 20%. But the level of improvement reached after injection 7 was 55%. Of course, such a difference in improvement is noticed by the patient. And it is achieved by the rigid regimen keeping cycle durations fixed to week 12 or week 13 and was lost when the rigid regimen had to be given up due to the Covid19-lock-down.

At the relevant places in the text we refer to Fig. 1 now.

Several years ago a doctoral student of HH has compared different aspects (side-affects, coherence to therapy) of the rigid regimen with treatment of patients with variable cycle durations.

Unfortunately, this thesis is available only in German so far. Nevertheless, we mention this thesis now in the references. 

Reviewer 2 Report

Authors address most of the concerns raised by us in the last review. However, there are minor corrections needed.

1) Fig 1a: It is difficult to differentiate red and green circles. So, please make it red and blue or green and blue circles.

2) Conclusion should be more elaborate and authors should discuss the effect of delay and possible strategy to avoid those.

Author Response

We have changed the colors green before lock-down and to blue after lock-down.

This point is well-taken:

Indeed, on the basis of the present data we have applied for an exceptional permission not to close the BoNT department but to continue the injection therapy.

This is mentioned in the text now.

Reviewer 3 Report

General and specific comments

This opportunistic manuscript (taking advantage of opportunities as they arise), in this case SARS-CoV-2-pandemic lockdown, deals with the administration of botulinum toxin to outpatient suffering from neurologic diseases in a German clinic.

The main problems with this manuscript are indicated here below:

  1. The number of patients for each of the disease entities investigated is too small
  2. Authors have pooled the data from the different diseases and have not indicated how is BoNT/A acting in those different diseases
  3. It is not clear also why some of the clinical conditions investigated need repeated injections
  4. Some statements need revision since it is not clear what the authors mean in the abstract and Introduction section by:

“Botulinum neurotoxin type A (BoNT/A) injections have to be performed repeatedly to achieve a permanent improvement”.

  1. It is clear that blepharospasm and facial hemispasm (FD), idiopathic cervical dystonia (CD), other dystonia as oromandibular, oropharyngeal, trunk und extremity dystonia (ODT), spasticity after infarct or trauma (SPAS) and pain syndromes, as migraine or tension headache (PAIN) do not obey to simple or unique physio pathological bases. Furthermore, it is well known that BoNT/A is a symptomatic treatment for most of these evolving diseases, and therefore BoNT/A can be conceive also as an evolving treatment. We know that some patients become refractory, or no-responder to the action of BoNT/A. Therefore, the term “permanent” improvement is not adequate; improvement is markedly dependent on the evolution of the mentioned diseases.
  2. Line 156, what are the bases of the the so called staircase-like continuous improvement until a stable level of improvement is reached in a patient with PAIN as compared to a patient with blepharospasm?
  3. Minor points line 71, spelling "und" should be replaced by "and"
  4. line 121, What represents (r= .66,
  5. Finally, it is unclear on the clinical conditions studied how the small delay leads to the worsening of symptoms in conditions so different as pain and movement disorders.

Author Response

Thanks for your valuable comments and time

1-Our question is: too small for what particular reason. This is just a short communication not an original work and original study is running at the time.

2-By far the number of patients was also large enough to demonstrate a significant worsening with a delay of 3 to 4 weeks.

3-It is unclear for us, that why should we indicate how BoNT/A acts in classical indications for BoNT/A?

4-All of these classical indications including pain syndromes need repeated injections. See Simpson et al. 2016 (reference 1).

5-This point is well-taken:

This sentence has been modified.

Reviewer 3 is right: permanent is not adequate. This has been modified.

Secondary non-response is a special research topic of our team. We did not want to touch this topic in the present paper.

The topic of disease progression has also recently been addressed by our team. But this slowly type of disease evolvement has nothing to do with rapid worsening after delayed reinjection. 

##Hefter H, Schomaecker I, Schomaecker M, Samadzadeh S. Disease Progression of Idiopathic Cervical Dystonia in Spite of Improvement After Botulinum Toxin Therapy. Front Neurol. 2020 Nov 12;11:588395. doi: 10.3389/fneur.2020.588395. PMID: 33281726; PMCID: PMC7689059.

6-The basis is daily experience in our outpatient clinic which is clearly demonstrated in Fig. 1. Patients rate severity of their disease in percent of the severity at onset of BoNT therapy. This rating can be done for all disease entities.

This is mentioned in the text more explicitly now.

7-Is corrected!

8-The non-parametric Spearman´s rho rank correlation coefficient. This is mentioned now!

9-The background is the following:

All these different disease entities are treated with BoNT. BoNTs are universal exocytosis blockers. The duration of BoNT depends on the presence of the light chain in the target cell independent on disease entity. With increase of duration reinjection cycle BoNT action inevitably declines independent on disease entity. 

Reviewer 4 Report

The manuscript simply presents the effect of SARS-CoV-2 pandemic lockdown and related reinjection delay on outcomes in patients of the clinic. The authors showed that small delay of just few weeks can lead to worsening of symptoms and relapse on a level of severity which therefore may take several treatment cycles to return to the prior stable benefit level before SARS-CoV-2 pandemic lockdown.

The manuscript is important in the field for both medical doctors and patients. I suggest to publish it as a correspondence without any modifications.

Author Response

Reviewer 4 picks-up and summarizes the main message of our paper very well. 

Thanks for your comments and time

Round 2

Reviewer 3 Report

The short communication, as stated in the author’s response to referees, is more exactly a commentary or correspondence written to express an opinion based on the impact of SARS-CoV-2-pandemic lockdown on botulinum toxin outpatient from a clinic in Germany.

Authors have corrected and complete some of the points raised previously. With  the corrections made by the authors at least erroneus concepts are not present in the commentary.

In relation to the response to the referees given as follows: ” All these different disease entities are treated with BoNT. BoNTs are universal exocytosis blockers. The duration of BoNT depends on the presence of the light chain in the target cell independent on disease entity. With increase of duration reinjection cycle BoNT action inevitably declines independent on disease entity.

I agree with most of what is indicated. However, other points that are not mentioned, is that duration depends on the concentration of BoNT/A injected, the type of nerve terminal blocked (motor, sensory, peptidergic, cholinergic), and the recovery rate of the protein cleaved by BoNT/A, in this case SNAP-25. Based on this, BoNT action is not independent of the disease entity.

This manuscript is a resubmission of an earlier submission. The following is a list of the peer review reports and author responses from that submission.

Round 1

Reviewer 1 Report

TSUI score etc should be written in full on first use, ideally in the abstract as the abbreviations make this almost incomprehensible

Was the "mean worsening" also in the TSUI score?

TSUI score (ATUSI-PTSUI) - also needs explanation 

Having the methods section at the end of the paper is somewhat unconventional - perhaps this should be inserted after the introduction?

the word weeks is missing in line 104

Patients being used to treatment with variable inter-injection intervals will be less sensitive to 111 the experience of worsening when the injection interval is prolonged - what is the evidence for this?  REF?

The statement - "Authors should discuss the results and how they can be interpreted in perspective of previous studies and of the working hypotheses. The findings and their implications should be discussed in the broadest context possible" - perhaps the authors here might discuss the potential hypotheses and implications of their findings

Author Response

Thank you so much for your time and comments.

1-TSUI score etc should be written in full on first use, ideally in the abstract as the abbreviations make this almost incomprehensible

A: TSUI score is the scoring system developed by Tsui comprises a rating for sustained movement amplitudes, duration, shoulder elevation and, in addition, for dystonic tremor. The Tsui score is not an abbreviation and presented by Tsui et al as cited in the manuscript.

2-Was the "mean worsening" also in the TSUI score?

A: The Tsui score is quantitative and the mean of changes was calculated and compared. (mean of PTSUI/ ATSUI/ ITSUI=4/5/5)

3-TSUI score (ATUSI-PTSUI) - also needs explanation 

A: It is explained in Materials and Methods section:

In patients with idiopathic CD, the TSUI-score at the present visit (ATSUI; [7]) was determined by the treating physician and by an independent physician not familiar with the treatment data of the patient (ITSUI). The TSUI-score determined at the previous injection visit was extracted from the charts (PTSUI).

The difference between ATSUI and PTSUI indicates worsening.

Physician´s rating of CD worsening by TSUI score (ATUSI-PTSUI) was significantly correlated with physician´s rating of worsening r = .43, p < .01 and was significantly correlated with PTSUI r = .41, p < .01. There was a highly significant correlation between the TSUI determined by the treating physician and the independent rater (ITSUI) r = .91, p < .001.

4-Having the methods section at the end of the paper is somewhat unconventional - perhaps this should be inserted after the introduction?

A: It is the Toxins journal format and style but it is corrected as your request to 1.introduction and then 2. Materials and Methods.

5-the word weeks is missing in line 104

A: Weeks is added in revised version

6-Patients being used to treatment with variable inter-injection intervals will be less sensitive to 111 the experience of worsening when the injection interval is prolonged - what is the evidence for this?  REF?

A: A patient who is used to the variability of outcome due to the variation of the inter-injection interval between 9 to 16 weeks or more will hardly notice a relevant worsening caused by a 4 weeks delay. That is our experience.

7-The statement - "Authors should discuss the results and how they can be interpreted in perspective of previous studies and of the working hypotheses. The findings and their implications should be discussed in the broadest context possible" - perhaps the authors here might discuss the potential hypotheses and implications of their findings

A:It is corrected in revised version.

Reviewer 2 Report

Suggestions are in PDF below.

Author Response

Thank you so much for your time and comments.

attached you could find the response to revision.

Reviewer 3 Report

This manuscript describes worsening of symptoms due to delayed toxin injection cause by early lockdown which in turn affect patient management. The observations and findings are interest to the people involved with botulinum toxin related therapeutics.

A few things should be improved in the manuscript.

  • Introduction should include dosing pattern (dose, frequencies and resistance, if any) in the mentioned disease areas.
  • I am not sure if the author modify the doses after the delay. Please clarify?
  • In Fig 1a, explain the two different color bulb. More explanation are required for the figure.
  • Fig 1b needs more elaborate description in the text.
  • Line 104, week?
  • Conclusion should be more scientific.

Author Response

Thanks for your time and comments.

1-Introduction should include dosing pattern (dose, frequencies and resistance, if any) in the mentioned disease areas.

A:The dosage of Botulinum Toxin is indicated in table 1. The interval and frequency were explained in text. (12 to 13 weeks)

The reviewer raises a very interesting point: the resistance to Botulinum toxin injections. This is a special topic of our team. We have performed the last cross-sectional study in our ambulance 2017, the presentation of these results is on the way. In 2017 these patients were negative in the MHDA-test and no patient developed an obvious secondary treatment failure since then. But we cannot be sure about the resistance if no tests have been performed. Therefore, we would not touch this topic in the present paper.

2-I am not sure if the author modify the doses after the delay. Please clarify?

A:We did not modify the dosage of Botulinum Toxin after the Delay since higher doses are associated with higher risk of ANTIBODY Development.

And it is explained in material and method section.

3-In Fig 1a, explain the two different color bulb. More explanation are required for the figure. Fig 1b needs more elaborate description in the text.

A:It is corrected:

Figure 1a.  one patient’s assessment of seven cycles before lockdown (green circles) and worsening during the lockdown(red circles)as an example of delay effect on worsening

Figure 1b. correlation between delay and worsening in the entire cohort.

4-Line 104, week?

A:Weeks is added in revised version

5-Conclusion should be more scientific.

A:Conclusion is changed according to the reviewer 1’s request.